# Non-Surgical Transversal Dentoalveolar Compensation with Completely Customized Lingual Appliances versus Surgically Assisted Rapid Palatal Expansion in Adults—Tipping or Translation in Posterior Crossbite Correction?

**DOI:** 10.3390/jpm13050807

**Published:** 2023-05-09

**Authors:** Jonas Q. Schmid, Elena Gerberding, Ariane Hohoff, Johannes Kleinheinz, Thomas Stamm, Claudius Middelberg

**Affiliations:** 1Department of Orthodontics, University of Münster, 48149 Münster, Germany; 2Department of Orthodontics, Hannover Medical School (MHH), 30625 Hannover, Germany; 3Private Practice, 49152 Bad Essen, Germany; 4Department of Cranio-Maxillofacial Surgery, University of Münster, 48149 Münster, Germany

**Keywords:** crossbite, surgically assisted rapid palatal expansion, surgically assisted rapid maxillary expansion, dentoalveolar compensation, expansion, mandibular constriction, mandibular compression, lingual orthodontics, buccolingual inclination, torque

## Abstract

The aim of this study was to investigate buccolingual tooth movements (tipping/translation) in surgical and nonsurgical posterior crossbite correction. A total of 43 patients (f/m 19/24; mean age 27.6 ± 9.5 years) treated with surgically assisted rapid palatal expansion (SARPE) and 38 patients (f/m 25/13; mean age 30.4 ± 12.9 years) treated with dentoalveolar compensation using completely customized lingual appliances (DC-CCLA) were retrospectively included. Inclination was measured on digital models at canines (C), second premolars (P2), first molars (M1), and second molars (M2) before (T_0_) and after (T_1_) crossbite correction. There was no statistically significant difference (*p* > 0.05) in absolute buccolingual inclination change between both groups, except for the upper C (*p* < 0.05), which were more tipped in the surgical group. Translation, i.e., bodily tooth movements that cannot be explained by pure uncontrolled tipping, could be observed with SARPE in the maxilla and with DC-CCLA in both jaws. Dentoalveolar transversal compensation with completely customized lingual appliances does not cause greater buccolingual tipping compared to SARPE.

## 1. Introduction

Treatment options for posterior crossbite in adults usually include surgically assisted rapid palatal expansion (SARPE), segmental osteotomy, microimplant-assisted rapid palatal expansion (MARPE) or dentoalveolar compensation.

To date, there is no consensus on the exact and most beneficial surgical procedure [1,2,3,4,5] or the hardware (tooth-borne/bone-borne) to be used with SARPE [6,7,8]. A recent systematic review showed an average expansion in the first molar region of 7.0 mm after SARPE, and since dental effects of SARPE were greater than skeletal ones, it was considered primarily a molar expansion procedure rather than a bodily skeletal expansion of the maxilla [9]. The decision to opt for surgically assisted rapid palatal expansion (SARPE) remains a subjective one, as there are no guidelines to select an age-appropriate approach to correct a transversal discrepancy [2].

Segmental osteotomy or two-piece maxilla provides the advantage of saving one surgical procedure. However, it is suitable for moderate transversal discrepancies [10,11] with intermolar expansion values from 2.7 to 4.7 mm [12,13], but there are problems with stability [14,15], with a relapse of up to 60% of the gained expansion [13].

MARPE is one of the invasive but not yet traumatic surgical methods. There is evidence that MARPE shows a parallel expansion pattern in young adults [16,17]. A recent systematic review in adult patients found a mean increase in skeletal width of 2.3 mm and a mean increase in intermolar width of 6.6 mm. Therefore, MARPE appears to have clinically comparable dental and skeletal effects to SARPE, but the evidence is still very limited [18].

Since SARPE is often only the prelude in patients undergoing orthognathic surgery, there is an ongoing quest to correct a transverse maxillo-mandibular discrepancy in a more patient friendly, i.e., less invasive way. This draws attention to transverse dentoalveolar compensation in both jaws as a possible therapeutic option for adults. It has been suggested that transverse discrepancies of up to 5 mm can be corrected by orthodontic tooth movement only [19,20], which has been confirmed by clinical studies focusing on the maxilla only [21,22]. Mainly, different types of maxillary expansion are performed for crossbite correction [23,24], whereas mandibular archform alteration is less common due to concerns about instability [25]. There are only a few case reports that show the option of dentoalveolar mandibular compression [26,27] or surgical constriction of the mandible [28].

Our first publication evaluating the same cohort of patients aimed to compare the amount of crossbite correction in adults treated either by SARPE and buccal straight wire appliances or non-surgically with transversal dentoalveolar compensation using completely customized lingual appliances (DC-CCLA) together with expansion and compression archwires [29]. Both concepts led to a comparable transverse correction in the posterior segment. However, the crossbite correction was achieved in different ways. Maxillary expansion was greater in the SARPE group due to the basal, surgically assisted expansion. Mandibular compression was greater in the DC-CCLA group [29] as the transverse compensation was realized in both jaws. Dentoalveolar compensation in the transverse dimension is frequently considered to have a greater risk of pure tooth tipping, whereas a surgically assisted concept is perceived to be more controlled, thus leading to fewer side-effects at the dentoalveolar level.

Therefore, the aim of this study was to compare the buccolingual inclination change and the type of tooth movements in adults with posterior crossbite treated either by SARPE or DC-CCLA. We tested against the null hypothesis that there is more tipping (change of buccolingual inclination) in the case of dentoalveolar compensation (DC) than with SARPE.

## 2. Materials and Methods

This is a follow-up study on DC-CCLA in comparison to SARPE. For the understanding of the work, the essential information is repeated here; specific details on the general study procedure can be found in the previous publication [29].

The study protocol was approved by the local Ethics Commission of the Medical Faculty of the University of Münster, Germany (2021-120-f-S). Two groups were formed to compare transversal crossbite correction in terms of buccolingual inclination change and translation of the posterior teeth: a surgical group treated with a surgically assisted rapid palatal expansion (SARPE) followed by a buccal straight wire appliance (SWA), and a DC-CCLA group treated with the WIN appliance (DW-Lingual Systems GmbH, Bad Essen, Germany) which consists of a completely customized lingual appliance (CCLA).

The source material for the measurements comprised digital models of consecutively debonded patients treated in a private practice (Bad Essen, Germany) during the period from 2019 to 2021 (DC-CCLA group) and of patients who underwent orthognathic surgery at the Department of Cranio-Maxillofacial Surgery, University Hospital Münster, Germany in the period from 2018 to 2021 (SARPE group). The measurements were taken at two timepoints: before treatment (T_0_) and at T_1_, which was after debonding in the DC-CCLA group or after alignment in both jaws prior to a second surgical intervention for three-dimensional bite correction in the SARPE group. At T_1_, crossbite correction was completed in both groups, and two situations comparable for evaluation were present [29]. In the first study, the linear metric correction in the transverse plane was analyzed using reproducible landmarks [29].

### 2.1. Measurement of the Buccolingual Inclination

The buccolingual inclination of the posterior teeth was measured in the transverse plane at T_0_ and T_1_. Stereolithography (STL) files of the models were imported into Meshmixer software (Autodesk, Inc., San Rafael, CA, USA), which was used to perform all measurements. The STL files were carefully aligned in the coordinate system of the software and superimposed, analogous to the previous publication [29]. After superimposition, the T_1_ STL file was then transformed along the *x*-axis so that an optimal view on the corresponding tooth surfaces of both models (T_0_ and T_1_) for setting of the landmarks was possible. According to Andrews [30], crown inclination refers to the labiolingual or buccolingual inclination of the long axis of the crown and is determined by the resulting angle between a line 90° to the occlusal plane and a line tangent to the middle of the labial or buccal clinical crown (also known as facial axis point). Therefore, at each canine (C), second premolar (P2), first molar (M1), and second molar (M2), two landmarks (one gingival and one occlusal) were placed in the vertical plane (long axis of the crown) passing through the facial axis point, e.g., geometric center of the vestibular crown surface (Figure 1 and Figure 2). The landmarks were placed along the long axis of the crown with an equal distance to the facial axis point to facilitate reproducibility. A straight line through the two landmarks of the respective side was constructed. The straight lines of each side intersect at the angle γ, which was measured at T_0_ and T_1_ to calculate the inclination change.

In case of insufficient representation of the tooth surfaces (e.g., molar bands), inclination was calculated over the occlusal surface. For this purpose, three reproducible landmarks were set, which computationally formed a plane (Figure 3). A calculated perpendicular through the center of the plane was used for the angular measurement.

### 2.2. Analysis of Tooth Movements: Uncontrolled Tipping or Bodily Movements

For a precise determination of the tooth movements that occurred in the posterior segment, the linear values of the tooth movement in the transversal dimension measured in the first study [29] were combined with the angular measurements obtained in the present study. Assuming pure uncontrolled tipping, the resulting linear change in the transverse dimension (expansion or compression) can be easily calculated if the position of the center of resistance is known. These values were compared with the actually measured linear values to obtain an indication of the extent of the bodily tooth movement that occurred.

The CR of a tooth is the point at which the tooth is most resistant to forces that are applied to it. Only in the two-dimensional model can this single point be calculated [31]. To answer the question whether the transversal change of the posterior teeth has taken place solely by uncontrolled tipping (rotation around CR) or by bodily tooth movements (translation), only the two-dimensional determination of CR is used hereafter.

The distance from the cusp tip to the center of resistance is, thus, one necessary distance for calculating the theoretical inclination with known transverse movement of the cusp tip. Different distance measurements to CR are given in the literature: starting from the geometric center of the crown [32], from the alveolar crest [33,34,35], and from the root apex [36]. From these data and from measurements of general crown length [37,38] and root lengths [39,40], the following average distances between cusp tip and CR were calculated with the consideration of the biological width [41]: canine (C) = 15.6 mm, second premolar (P2) = 13.7 mm, first molar (M1) = 12.5 mm, and second molar (M2) = 11.5 mm. Knowledge of the linear and angular change allows the determination of the tooth movement that has occurred (Figure 4).

The tooth is assumed to rotate in CR with the cusp tip moving to buccal or lingual by distance b. Individual measurements of distance b were taken from the previous publication [29]. The distance c is composed of the distance from CR to the alveolar crest plus the biological width plus the crown height. If the known distance b is greater than the distance b calculated on the basis of pure rotation in CR, it can be assumed that additional translation has taken place. A greater known distance b and a smaller inclination change (γ) correlate with greater root movement.

### 2.3. Statistical Analysis

To measure the relationship between the respective intervention (either SARPE or DC-CCLA) and gender, as well as between intervention and jaw relation (i.e., Angle class), a chi-square test was used. To determine if there were differences in the change of buccolingual inclination between the SARPE and DC-CCLA group, the Mann–Whitney U-test was used. A Wilcoxon signed-rank test was applied to examine group differences between the known distance b and the calculated distance b in case of pure rotation. The significance level was set to α = 5%, and a *p*-value < 0.05 was considered significant. No α-correction for multiple testing was performed due to the exploratory nature of the study. All statistics were performed using the software SPSS Statistics 29 for Mac (IBM Corp., Armonk, NY, USA).

The method error was assessed by intraindividual reproducibility of the measured angles. For this purpose, the principal investigator (JQS) measured 10 randomly selected models at two different timepoints. The measurement error was determined using Dahlberg’s formula [42].

## 3. Results

The SARPE group (n = 43, mean age 27.6 ± 9.5 years) consisted of 19 female (mean age 26.7 ± 9.7 years) and 24 male (mean age 27.8 ± 9.8 years) patients. The DC-CCLA group (n = 38, mean age 30.4 ± 12.9 years) included 25 females (mean age 32.1 ± 12.3 years) and 13 males (mean age 27.0 ± 14.2 years). The baseline characteristics are summarized in Table 1. The two groups did neither differ in age at the beginning of treatment (T_0_, *p* > 0.05) nor at its end (T_1_, *p* > 0.05). No significant difference was found between gender and intervention (*p* > 0.05), but there was a statistically significant difference for intervention and Angle classification (*p* < 0.001). According to Dahlberg’s formula, a measurement error of 2.16° must be assumed for this study.

### 3.1. Buccolingual Inclination Change in the Maxilla

Although the transverse deficiency in the maxilla was entirely corrected by tooth movements in the DC-CCLA group, the amount of mean buccal or lingual tipping was less than 11.6° (Figure 5 and Table 2) and not significantly different between the two groups at P2 (*p* > 0.05), M1 (*p* > 0.05), and M2 (*p* > 0.05). In the canine area, significantly more tipping occurred in the SARPE group (*p* < 0.05). The maximum inclination change at the four locations was between 25.3° and 41.1° in the SARPE group. Comparable values for the DC-CCLA group were slightly smaller (16.4° to 35.5°).

Figure 6 and Table 3 show the direction (buccal/lingual) in which the tipping occurred. In the SARPE group, the canines were tipped lingually significantly. In the DC-CCLA group, the greatest buccal tipping was found at P2, with small values observed at M1 (Table 3), although significant expansion of 3.7 mm [29] was achieved here. Despite a small maxillary expansion of 0.4 mm [29] a lingual crown tipping was observed at M2 in the DC-CCLA group. Overall, there was a statistically significant difference in maxillary inclination change between the SARPE and DC-CCLA groups at C (*p* < 0.05), P2 (*p* < 0.05), and M2 (*p* < 0.05). There was no statistically significant difference at the first molars (*p* > 0.05).

### 3.2. Buccolingual Inclination Change in the Mandible

Although the entire transverse deficiency in the posterior region was partially corrected by mandibular compression in the DC-CCLA group, the average amount of buccal or lingual tipping was less than 10.7° (Figure 5 and Table 2) and similar to the SARPE group. In detail, there was no statistically significant difference in mandibular absolute inclination change between the SARPE and DC-CCLA groups at C (*p* > 0.05), P2 (*p* > 0.05), M1 (*p* > 0.05), and M2 (*p* > 0.05). The maximum inclination change at the four locations was between 18.7° and 37.7° in the SARPE group. Comparable values for the DC-CCLA group were slightly higher (26.0° to 44.0°).

The direction in which the tipping took place is illustrated in Figure 6 and Table 3. In the mandible, significant differences between the two groups could be noted only in the area of the second molar. In the SARPE group, buccal tipping of the second molar was found. Due to the compression of −3.5 mm [29] a comparable amount of tipping to the lingual side could be seen in the DC-CCLA group.

Overall, there was a statistically significant difference in the change in mandibular inclination between the SARPE and DC-CCLA groups only at M2 (*p* < 0.001). There was no statistically significant difference at C (*p* > 0.05), P2 (*p* > 0.05), and M1 (*p* > 0.05).

### 3.3. Description of Tooth Movements in the Maxilla

In the case of pure rotation in CR (Figure 4a), the following theoretical values per 1 mm expansion result, taking into account the calculated distances from the cusp tip to CR: C = 3.7°, P2 = 4.2°, M1 = 4.6°, and M2 = 5.0°. This linear relationship between rotation and displacement of the cusp tip (expansion) can be compared with the expansion values from the previous publication [29]. The scatterplots in Figure 7 show which inclination measurements correspond to an uncontrolled tipping and which do not, i.e., must have an additional translation of the center of resistance (Figure 4b,c).

All measurements of the maxilla in the SARPE group (except one canine in a buccal position) showed a combination of tipping and translation to the buccal side. However, the values were inconsistent in relation to the tooth groups; for example, the canines showed both large constriction and large expansion values. Movements of all tooth groups were significantly different (*p* < 0.001) from pure tipping.

In the DC-CCLA group, the majority of cases also showed tooth movements that were a combination of tipping and translation, indicating an overall controlled tooth movement. The individual tooth groups were more uniformly distributed; for example, no extreme values were found for canine movements, but only values around the zero point. Here, the individual tooth groups also differed significantly (*p* < 0.001) in their movement from pure tipping, except for the canines (*p* > 0.05).

### 3.4. Description of Tooth Movements in the Mandible

In the mandible of the SARPE group, the measurement points were grouped around the diagonal without a pattern, indicating more or less uncontrolled tipping (Figure 7, lower left). It is important to note that many cases were expanded in the M1 and M2 region, making the crossbite correction more difficult. Movements of all tooth groups were not significantly different from uncontrolled tipping (*p* > 0.05).

Looking at the DC-CCLA group in the lower jaw (Figure 7, lower right), it is noticeable that most of the measuring points were to the left of the diagonal line, indicating a targeted translation to the lingual side and a rather controlled tooth movement. The majority of the measuring points of M1 and M2 were in the lower negative quadrant of the scatterplot, which indicates lingual translation with pronounced lingual root movement. Movements of P2, M1, and M2 were significantly different (*p* < 0.001) from uncontrolled tipping.

## 4. Discussion

The present study investigated the inclination change of posterior teeth with crossbite correction in two groups: in a surgical SARPE group and in a nonsurgical DC-CCLA group. In a previous study on identical patients [29], it was shown that dentoalveolar compensation can achieve the same total crossbite correction as SARPE. The question now arose as to whether the correction took place only by uncontrolled tipping of the teeth or also by translation.

Since the risk of side-effects must be considered similar for lingual or buccal tipping, absolute values of the inclination change were calculated. The results of the present study showed comparable absolute tipping values (buccal or lingual) for both treatment modalities. There was no statistically significant difference in absolute buccolingual inclination change between both groups, except for the upper canines, which showed more tipping in the SARPE group.

In the maxilla, buccal tipping was greater in the DC-CCLA group at C and P2, while buccal tipping—despite significant dentoalveolar expansion—was similar at M1, and controlled root torque occurred at M2 in the DC-CCLA group. In the mandible, there was no statistically significant difference in lingual tipping between the groups, except for M2. In the SARPE group, the lower second molars were even tipped buccally. Bodily tooth movements that cannot be explained by pure uncontrolled tipping could be observed with dentoalveolar compensation in both jaws except for the canines.

It should be noted that a clinically significant amount of tipping occurred in both groups, with no significant differences between the groups, except for the upper canines, which showed more tipping in the SARPE group. A possible explanation would be that a certain amount of crown tipping is unavoidable with SARPE and the tipping values for dentoalveolar compensation were smaller than expected because the lower arch was included in the correction of the crossbite. Therefore, the corrective movements were somehow distributed among the four quadrants. Handelman et al. showed that nonsurgical rapid maxillary expansion in adults resulted in expansion values of 4.6 mm and 6.2° buccal tipping at M1 [21]. The fact that the values of the DC-CCLA group in our study were smaller with 2.2° buccal tipping at the upper M1 could be explained by the fact that more translation took place. In addition, the expansion was carried out slowly, which was later also recommended by Handelman [43]. However, it must be emphasized that some amount of buccal tipping of the first molars also occurs with SARPE. Buccal tipping at M1 was found to be 3.1° with tooth-bone-borne appliances and 3.8° with tooth-borne-appliances six months after SARPE with expansion values of 6.2/6.8 mm [44]. An inclination change to buccal of 2–4° was confirmed by Ferraro-Bezzera and coworkers [5]. Our results showed lower tipping values of 0.3° at M1 with SARPE, which were within the measurement error and indicate a bodily expansion of the bony pieces in the maxilla. It is also noticeable that the maxillary canines in the SARPE group were significantly tipped lingually. A possible explanation would be that there is a large expansion in the anterior region with SARPE [45,46], and lingual tipping of the canines occurs as part of the anterior space closure. Furthermore, a clinically significant amount of buccal tipping was apparent at the lower second molars. Buccal tipping and additional expansion in the mandible must be considered contraindicated as it complicates the crossbite correction and may require even greater expansion in the maxilla. One possible explanation would be the dependence on the arch form that was used with the SWA. It has been shown that preformed archwires are often too broad for patients in the intercanine and intermolar region [47,48], and different degrees of tipping (Figure 7) in the mandible could be explained with an unavoidable torque play of metal injection-molded brackets [49].

The most important aspect in the nonsurgical approach is the fact that the lower arch was included in the crossbite correction. A change in the mandibular archform is less common due to concerns about instability [25]. However, it has been found that the more the archform is altered during treatment, the greater the tendency is for relapse [25,50], which supports the concept of not only correcting the crossbite in one jaw, but also distributing the correction to all four quadrants. Correction of the crossbite in both jaws is not a new concept. It has already been hypothesized in the past that, if the mandibular arch is too wide, correction of a maxillo-mandibular discrepancy by altering the upper arch alone carries a high risk of failure; therefore, treatment should include both arches [51,52]. Compression of the arch form in the mandible is not common and has not yet been scientifically investigated, but would in fact be more in line with the natural tendency of the lower arch to become narrower with age when no form of retention is applied [53]. Clinical examples of nonsurgical dentoalveolar compensation of posterior crossbites in both jaws can be seen in Figure 8.

A major finding of this study is that translational tooth movements with dentoalveolar compensation in the transverse dimension are possible. It is often doubted that the alveolar process allows sufficient bodily tooth movement in the transverse direction. Pure tipping in the buccolingual direction (rotation around CR) is an uncontrolled tooth movement and seems to be associated with the risk of inadequate tooth position and poor occlusion, as well as the occurrence of dehiscences [54]. Tipping leads to compressive forces in the occlusal and apical third of the root [55] and could lead to alveolar bone loss due to the high strains acting on the root [56]. In theory, distributing the force over the entire root surface as with translation of the root in the direction of the cortex could reduce the side-effects of tooth movements in the transverse dimension [57]. Capps and coworkers showed that buccal bone apposition is possible in buccal bodily tooth movement when a suitable force system is used [57]. In this case, not only medullar bone but also cortical bone was apparently newly formed [57]. Our results support the theory that bone formation is possible with transversal tooth movement. Otherwise, a significant amount of translation with maximum values of 5–7 mm in the DC-CCLA group (Figure 7) would not have been possible. This buccolingual dentoalveolar movement seems comparable to the sagittal dentoalveolar effect of the Herbst treatment in adults. It was shown that, at the end of the Herbst/multibracket treatment, the lower incisors were translated on average 2.7 mm [58] and 2.1/2.4 mm [59] anteriorly together with their alveolar bone without significant periodontal damage. This means that medullar and cortical bone has apparently been newly formed. In the present study, the mean translation in the DC-CCLA group was 2–3 mm (Figure 7) and, therefore, comparable to the sagittal Herbst effect. The hypothesis, thus, seems logical that the same compensatory mechanisms can be attributed to the alveolar process in both the transverse and the sagittal direction.

### Strength and Limitations of the Study

Because subjects, recruitment, and models did not change from the previous study [29], identical limitations can be assumed. These include the retrospective design and the group differences in the Angle classification. One criticism concerns the different orthodontic referrers of surgery patients, who may have used different archwire shapes, thus influencing the results. The archwires in the DC-CCLA group were also individualized depending on the initial malocclusion. All patients in the SARPE group received a second surgical intervention for three-dimensional bite correction, which explains a higher number of class III and fewer class I cases.

The design of this study did not allow conclusions about stability or other side-effects such as gingival recessions. Further studies are necessary to clarify the open questions about long-term stability and possible side-effects such as periodontal problems of both treatment protocols.

The strength of this study can be seen in the high number of included patients, particularly in the DC-CCLA group, strengthening the evidence for dentoalveolar compensation. Another strength is the three-dimensional measurement process, which provides information about crown and root inclinations that could otherwise only be evaluated by invasive methods such as CBCT. The latter would not be feasible on purely orthodontic patients for medical and ethical reasons.

## 5. Conclusions

Dentoalveolar transversal compensation with completely customized lingual appliances does not cause greater buccolingual tipping compared to SARPE. Bodily tooth movements together with the surrounding alveolar process are possible with SARPE in the maxilla. Bodily tooth movements in the transverse dimension are possible with CCLAs in both jaws.

## Figures and Tables

**Figure 1 jpm-13-00807-f001:**
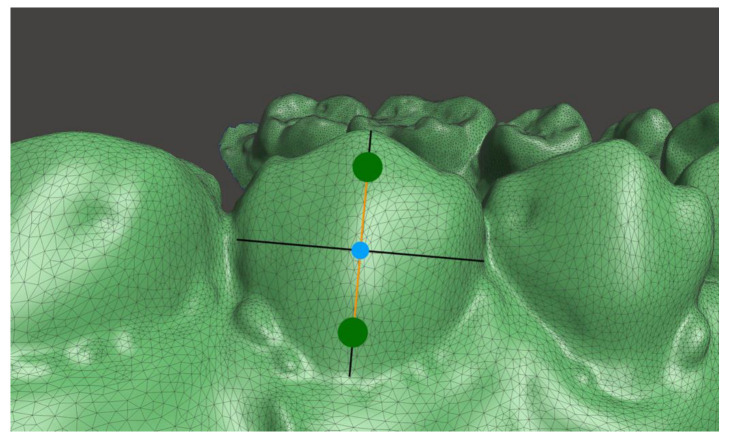
Schematic representation of setting the landmarks on a lower second premolar (P2). Two landmarks (green points) were placed in the Meshmixer software (Autodesk, Inc., San Rafael, CA, USA) on the buccal surface along the long axis of the crown passing through the facial axis point (blue point) defined as the geometric center of the vestibular crown surface (black lines) These landmarks were placed with an equal distance (orange line) to the facial axis point to facilitate reproducibility between T_0_ and T_1_.

**Figure 2 jpm-13-00807-f002:**
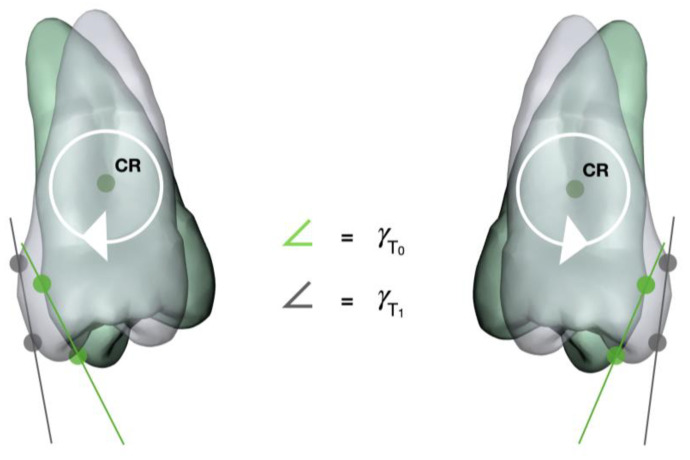
Schematic representation of measuring the inclination of upper second premolars. Two landmarks (gingival and occlusal) were placed on the crown surface of the digital models at T_0_ (green) and T_1_ (gray). In this example, a buccal crown tipping/lingual root tipping is shown with the teeth rotating in their center of resistance (CR). The connections of the landmarks per side form an angle γ. The difference in γ between T_0_ and T_1_ is the change in inclination that occurred during treatment.

**Figure 3 jpm-13-00807-f003:**
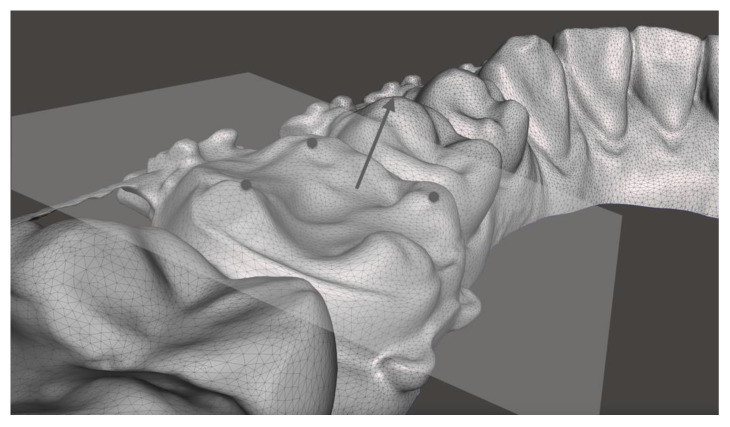
Schematic representation of setting the landmarks on a lower first molar (M1) with an orthodontic band. Three landmarks (gray points) were placed in the Meshmixer software on the occlusal surface to computationally form a plane. The landmarks were placed on specific locations on the cusps of the molar that were as reproducible as possible between T_0_ and T_1_. The inclination was measured in the frontal plane between the normal vectors (gray arrow) of both sides.

**Figure 4 jpm-13-00807-f004:**
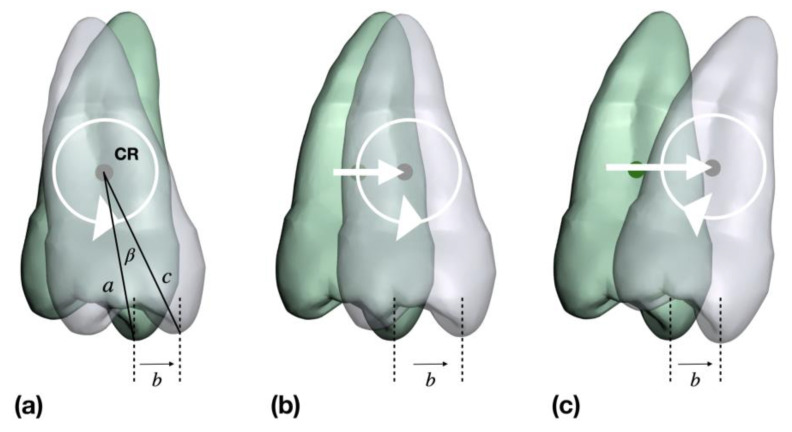
Examples of tooth movements based on the measured inclination and buccal movement of the cusp tip of tooth 25. Green indicates the situation at T_0_ and gray indicates the situation at T_1_. Distances of *b* were taken from the previous publication [29]. (**a**) The measured buccal movement *b* can be explained by a pure rotation in the center of resistance (CR) around the measured angle β, which is determined by the line a connecting the cusp tip at T_0_ with CR and the line c connecting the cusp tip at T_1_ with CR. (**b**) The measured buccal movement *b* cannot be explained by a pure rotation in CR, since the inclination change is smaller than it would be in pure rotation. The buccal movement can, therefore, be explained by an additional translation of the tooth. (**c**) In this case, the inclination change is negative, and the tooth should have undergone a rotation toward lingual in case of pure rotation in CR. Nevertheless, a buccal movement has taken place, which can, therefore, be explained by translation with a pronounced root movement to buccal.

**Figure 5 jpm-13-00807-f005:**
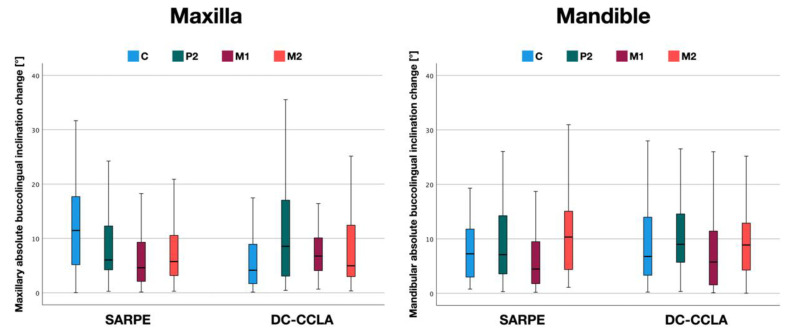
Buccolingual inclination change in the maxilla and the mandible to either side (absolute values) with surgically assisted rapid palatal expansion (SARPE) and dentoalveolar compensation with completely customized lingual appliances (DC-CCLA), measured in the region of the canine (C), second premolar (P2), first molar (M1), and second molar (M2).

**Figure 6 jpm-13-00807-f006:**
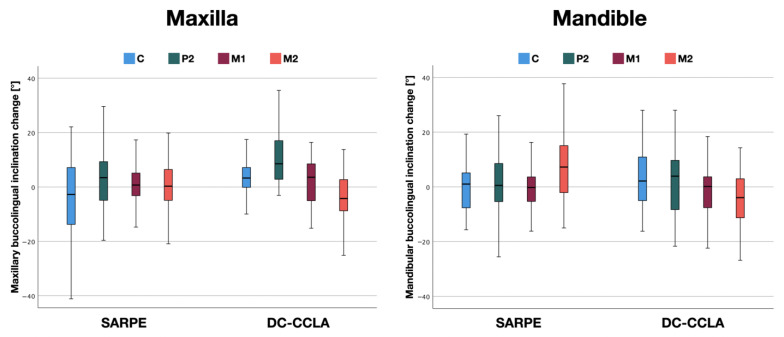
Buccolingual inclination change in the maxilla and the mandible positive values = inclination change to buccal, negative values = inclination change to lingual) with surgically assisted rapid palatal expansion (SARPE) and dentoalveolar compensation with completely customized lingual appliances (DC-CCLA).

**Figure 7 jpm-13-00807-f007:**
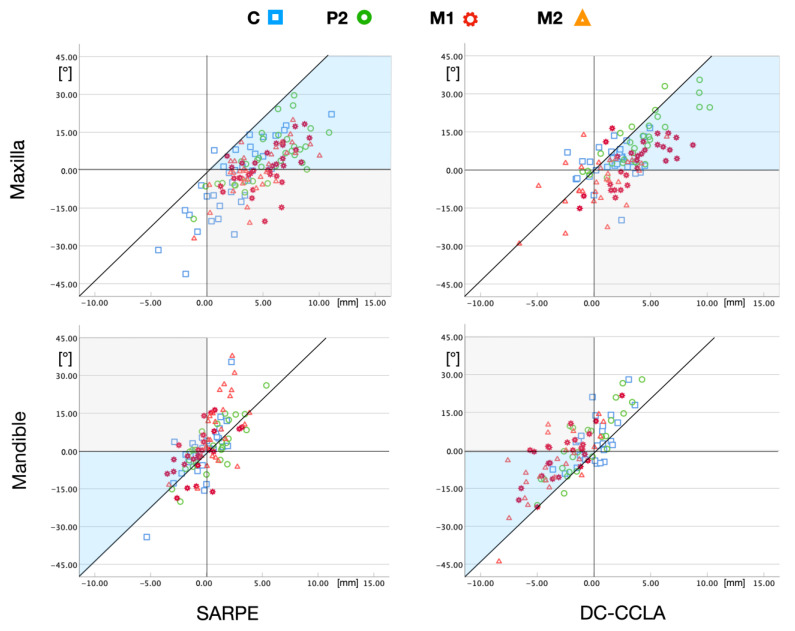
Buccolingual inclination change (*y*-axis) and buccolingual movement (*x*-axis) of the cusp tips (corresponds to distance b in Figure 4). A positive inclination change means crown tipping to buccal and vice versa. The same definition applies to the buccolingual movement, whereby positive values mean buccal movement and vice versa. The diagonal line represents a transverse movement of the cusp tip (expansion/constriction) that only occurs due to uncontrolled tipping (rotation in CR). Measuring points on the left side of the diagonal line denote a movement that is not only caused by pure rotation but also by a translation to lingual. The further the measuring point is to the left of the diagonal, the greater the lingual translation or the respective root movement. Similar applies to the right side of the diagonal. These measuring points can be explained by additional translation to buccal. The further the measuring point is from the diagonal, the greater the buccal translation or the respectiveroot movement. Measuring points in the blue field indicate movements shown in Figure 4b, and measuring points in the gray field indicate movements shown in Figure 4c.

**Figure 8 jpm-13-00807-f008:**
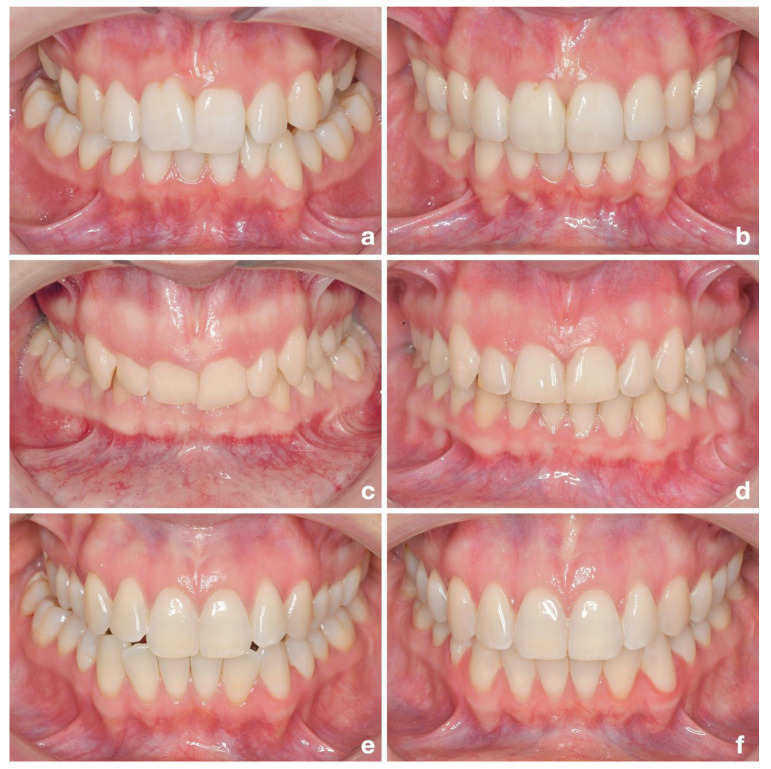
Clinical examples of nonsurgical dentoalveolar compensation (DC) of posterior crossbites with CCLAs. (**a**) Adult patient with bilateral posterior crossbite and crowding in both jaws. (**b**) After DC-CCLA treatment with expansion and compression stainless-steel archwires. (**c**) Adult patient with bilateral crossbite and a class II occlusion with frontal deep bite. (**d**) Situation after class II correction and bilateral crossbite correction. (**e**) Adult patient with unilateral crossbite. (**f**) After crossbite correction with expansion and compression archwires, the midline is centered.

**Table 1 jpm-13-00807-t001:** Baseline characteristics (mean ± SD) of the groups concerning intervention, gender, Angle class, and age in years.

	SARPE	DC-CCLA	*p*
Female	19 (44%)	25 (66%)	
Age (years) at T_0_	26.7 ± 9.7	32.1 ± 12.3	0.086
Age (years) at T_1_	30.1 ± 9.6	34.4 ± 12.4	0.255
Male	24 (56%)	13 (34%)	
Age (years) at T_0_	27.8 ± 9.8	27.0 ± 14.2	0.404
Age (years) at T_1_	30.9 ± 9.5	30.0 ± 14.4	0.276
Angle class I	1	20	
Angle class II	14	13	
Angle class III	28	5	

**Table 2 jpm-13-00807-t002:** Mean absolute values (M) of tipping to both sides (buccal and lingual), standard deviations (SD) and 95% confidence intervals (CI), minimum (Min) and maximum (Max) absolute values (°), and *p*-values for maxillary (Mx) and mandibular (Md) inclination change.

Jaw	Tooth	SARPE						DC-CCLA					*p*
		M	SD	95% CI	Min	Max		M	SD	95% CI	Min	Max	
Mx	C	12.21	9.40	9.21–15.21	0.03	41.13		6.40	5.91	4.45–8.34	0.12	23.74	0.005
	P2	8.79	7.16	6.47–11.11	0.28	29.65		11.60	9.99	8.06–15.14	0.44	35.53	0.416
	M1	6.50	6.11	4.52–8.48	0.13	25.31		7.33	4.27	5.86–8.80	0.67	16.41	0.156
	M2	7.47	6.34	5.41–9.52	0.29	27.09		8.25	7.45	5.61–10.90	0.35	29.07	0.804
Md	C	8.24	7.69	5.81–10.67	0.78	35.37		8.57	6.76	6.35–10.80	0.22	27.99	0.573
	P2	8.68	6.78	6.35–11.01	0.30	26.03		10.67	7.45	7.84–13.50	0.35	28.02	0.202
	M1	6.45	5.72	4.45–8.45	0.20	18.70		8.18	7.64	5.22–11.15	0.12	25.99	0.515
	M2	11.74	9.11	8.57–14.93	1.11	37.73		10.47	9.23	7.09–13.85	0.03	43.96	0.478

**Table 3 jpm-13-00807-t003:** Mean values (M), standard deviations (SD), and 95% confidence intervals (CI), minimum (Min) and maximum (Max) values (°), and *p*-values for maxillary (Mx) and mandibular (Md) inclination change.

Jaw	Tooth	SARPE						DC-CCLA					*p*
		M	SD	95% CI	Min	Max		M	SD	95% CI	Min	Max	
Mx	C	−4.47	14.86	−9.22–0.28	−41.13	22.14		3.99	7.78	1.43–6.55	−19.80	23.74	0.006
	P2	3.59	10.83	0.08–7.10	−19.63	29.65		11.13	10.52	7.40–14.86	−3.07	35.53	0.007
	M1	0.28	8.98	−2.63–3.19	−25.31	18.25		2.20	8.28	−0.64–5.05	−15.16	16.41	0.398
	M2	−0.07	9.87	−3.26–3.13	−27.09	19.84		−4.94	10.03	−8.50–−1.39	−29.07	13.79	0.029
Md	C	−0.59	11.33	−4.16–2.99	−34.17	35.37		2.99	10.59	−0.49–6.46	−16.20	27.99	0.185
	P2	1.01	11.07	−2.79–4.81	−25.55	26.03		3.06	12.80	−1.81–7.93	−21.68	28.02	0.651
	M1	−0.74	8.66	−3.77–2.28	−18.70	16.26		−0.41	11.30	−4.79–3.98	−22.39	25.99	0.944
	M2	7.92	12.67	3.50–12.34	−14.99	37.73		−4.86	13.19	−9.69–−0.02	−43.96	25.18	<0.001

Positive values = inclination change to buccal, negative values = inclination change to lingual.

## Data Availability

The data underlying this article can be shared upon reasonable request to the corresponding author.

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
