# Peer review of "Non-Surgical Transversal Dentoalveolar Compensation with Completely Customized Lingual Appliances versus Surgically Assisted Rapid Palatal Expansion in Adults—Tipping or Translation in Posterior Crossbite Correction?"

_jpm, 2023, doi:10.3390/jpm13050807_

Round 1

Reviewer 1 Report

I congratulate the Authors for their interesting study comparing dental movements (tipping/translation) after SARPE or customized lingual appliances. The subject is of high clinical interest and observational data are still limited.
The Authors have also to be praised for the methodological approach, i.e. taking advantage of scanned 3D occlusal models instead of CT scans.

Author Response

Dear Reviewer,

Thank you for taking the time to review our manuscript. Please find our answers in the attachment.

Sincerely,

Jonas Q. Schmid

Reviewer 2 Report

Hello,

My compliments for the article. I have read it with much interest since SARPE and surgical procedures are yet considered the top of dental position changes.

In Methods, lines 139-139, I would like a draw that better explains the molar landmarks and the angular measurement.

The English are good.

I have recommended it for publishing.

Author Response

(The authors gave the same response as above.)
